

**A growing threat to the ozone layer from short-lived anthropogenic**
**chlorocarbons**
David E. Oram (1,2), Matthew J. Ashfold (3), Johannes C. Laube (2), Lauren J. Gooch (2),
Stephen Humphrey (2), William T. Sturges (2), Emma Leedham-Elvidge (2,8), Grant L. Forster
(1,2), Neil R.P. Harris (4), Mohammed Iqbal Mead (4,5), Azizan Abu Samah (5), Siew Moi
Phang (5), Chang-Feng Ou-Yang (6), Neng-Huei Lin (6), Jia-Lin Wang (7), Angela K. Baker
(8), Carl A. M. Brenninkmeijer (8) and David Sherry (9)
(1) National Centre for Atmospheric Science, School of Environmental Sciences, University of
East Anglia, Norwich, NR4 7TJ, UK (d.e.oram@uea.ac.uk); (2) Centre for Ocean and
Atmospheric Sciences, School of Environmental Sciences, University of East Anglia, Norwich,
UK; (3) School of Biosciences, University of Nottingham Malaysia Campus, 43500 Semenyih,
Malaysia; (4) Centre for Atmospheric Informatics and Emissions Technology, School of
Energy, Environment and Agrifood/Environmental Technology, Cranfield University, UK; (5)
Institute of Ocean and Earth Sciences, University of Malaya, Kuala Lumpur, Malaysia; (6)
Department of Atmospheric Sciences, National Central University, Chungli, Taiwan; (7)
Department of Chemistry, National Central University, Chungli, Taiwan; (8) Max Planck
Institute for Chemistry, Air Chemistry Division, Mainz, Germany; (9) Nolan Sherry &
Associates, Kingston upon Thames, UK.
**Abstract**
Large and effective reductions in emissions of long-lived ozone-depleting substance (ODS)
are being achieved through the Montreal Protocol, the effectiveness of which can be seen in
the declining atmospheric abundances of many ODS. An important remaining uncertainty
concerns the role of very short lived substances (VSLS) which, owing to their relatively short
atmospheric lifetimes (less than 6 months), are not regulated under the Montreal Protocol.
Recent studies have found an unexplained increase in the global tropospheric abundance of
one VSLS, dichloromethane ($CH_2Cl_2$), which has increased by around 60% over the past
decade. Here we report dramatic enhancements of several chlorine-containing VSLS,
including $CH_2Cl_2$ and $CH_2ClCH_2Cl$ (1,2-dichloroethane), observed in surface and upper
tropospheric air in East and South East Asia. Surface observations were an order of
magnitude higher than previously reported in the marine boundary layer, whilst upper
tropospheric data were up to 3 times higher than expected. In addition we provide further
evidence of an atmospheric transport mechanism whereby substantial amounts of industrial
pollution from East Asia, including these chlorinated VSLS, can rapidly, and regularly, be
transported to tropical regions of the western Pacific and subsequently uplifted to the tropical
upper troposphere. This latter region is a major provider for air entering the stratosphere and
so this mechanism, in conjunction with increasing emissions of Cl-VSLS from East Asia, could
potentially slow the expected recovery of stratospheric ozone.
**1. Introduction**
Large-scale ozone depletion in the stratosphere is a persisting global environmental problem.
It is predominantly caused by the release of reactive chlorine and bromine species from
halogenated organic compounds. Although the basic science is well established, there
remains significant uncertainty surrounding the long-term recovery of the ozone layer (Hegglin
et al., 2015). One important issue is the recent, unexplained increase in the global tropospheric
abundance of dichloromethane ($CH_2Cl_2$), which has increased by ~60% over the past decade
(Leedham-Elvidge et al., 2015; Hossaini et al., 2015a; Carpenter and Reimann et al., 2015).





$CH_2Cl_2$ is one of a large group of halogenated compounds known as VSLS (very short-lived substances). Owing to their relatively short atmospheric lifetimes (typically less than 6 months) and their correspondingly low Ozone Depletion Potentials (ODPs), VSLS are not currently regulated by the Montreal Protocol. It is however estimated that a significant fraction of VSLS and their atmospheric degradation products reach the stratosphere (>80% in the case of chlorinated VSLS; Carpenter and Reimann et al., 2015) and, furthermore, halogenated VSLS have been shown to have a disproportionately large impact on radiative forcing and climate due to their atmospheric breakdown, and the subsequent depletion of ozone, occurring at lower, climate sensitive altitudes (Hossaini et al., 2015b). According to the most recent Scientific Assessment of Stratospheric Ozone Depletion (Carpenter and Reimann et al., 2015) over the period 2008-2012 the total chlorine from VSLS increased at a rate of approximately $1.3 \pm 0.2$ ppt Cl yr$^{-1}$, the majority of this increase being due to $CH_2Cl_2$, and this has already begun to offset the decline in total tropospheric chlorine loading over the same period ($13.4 \pm 0.9$ ppt Cl yr$^{-1}$) caused by the reduced emissions of substances controlled by the Montreal Protocol.

In recent years much attention has been focussed on the potential of bromine-containing VSLS to contribute to stratospheric ozone depletion (Law and Sturges, 2007; Montzka and Reimann, 2011). This is primarily due to the large observed discrepancy between the measured inorganic bromine in the stratosphere and the amount of bromine available from known, longer lived source gases, namely the halons and methyl bromide (Dorf et al., 2006). In contrast, the role of very short-lived chlorine compounds (Cl-VSLS) in ozone depletion has been considered relatively minor because they are believed to contribute only a few percent to the total chlorine input to the stratosphere, the majority of which is supplied by long-lived compounds such as the chlorofluorocarbons (CFCs), methyl chloroform ($CH_3CCl_3$) and carbon tetrachloride ($CCl_4$). Since 1987 the consumption of these long-lived anthropogenic compounds has been controlled by the Montreal Protocol and the sum of total organic chlorine in the troposphere has been falling since its peak of around 3660 parts per trillion (ppt) in 1993/94 to ~3300 ppt in 2012 (Carpenter and Reimann et al., 2015). Because of its relatively short atmospheric lifetime (~5 years) and its high chlorine content (3 chlorine atoms per molecule), the main contributor to this decline has been $CH_3CCl_3$. However, most $CH_3CCl_3$ has now been removed from the atmosphere with a present day abundance of less than 5 ppt. Consequently the rate of decline in total organic chlorine has fallen to 13.4 ppt/year (2008-2012), which is around 50% smaller than the maximum seen in the late 1990s (Carpenter and Reimann et al., 2015).

Owing to their short atmospheric lifetimes and their hitherto low background concentrations, chlorinated VSLS have not been considered of major importance for ozone depletion. Indeed the contribution of VSLS to the total chlorine entering the stratosphere is estimated to be only 55 (38–95) ppt (Carpenter and Reimann et al., 2015), which is between 1% and 3% of the present day (2012) total (3300 ppt). However, because of their short lifetimes, the potential impact of VSLS on stratospheric ozone is highly dependent on the location of their sources, with emissions close to the major stratospheric input regions being of far greater significance for ozone depletion.

The transport of trace gases and aerosols from the troposphere into the stratosphere occurs primarily in the tropics, where convective activity and vertical uplift are most intense. In order to get to the stratosphere an air parcel has to pass through the tropical tropopause layer (TTL),



the region of the atmosphere between the level of maximum convective outflow (~12 km
altitude, 345K potential temperature) and the cold-point tropopause (~17 km, 380K). The
vertical flux into the TTL is thought to be dominated by two main regional pathways, (1) ascent
above the western Pacific during Northern Hemispheric (NH) winter and (2) the circulation of
the Asian (Indian) Monsoon during NH summer (Fueglistaler et al., 2009). The latter has been
suggested as the most important region for transport of anthropogenic pollution (Randel et al.,
100   2010).

Because of their short lifetimes, to be able to accurately determine the VSLS contribution to
total organic halogen loading in the stratosphere it is highly desirable to collect data in the
TTL, surface measurements alone, particularly in regions outside the tropics where most long-
term surface stations are sited, are not sufficient. Furthermore, because of the distribution and
seasonality of stratospheric entry points it is also essential to measure in specific locations
and at specific times of year, i.e. in the Indian summer monsoon and over the winter western
Pacific. Unfortunately there are very few available measurements of VSLS in the TTL generally
as it is above the maximum altitude of most research aircraft, and, furthermore, there is a
paucity of both ground and aircraft data available in these two key regions of interest. Where
recent TTL data is available it is primarily from different regions and focussed on brominated
VSLS (e.g. Sala et al., 2014; Navarro et al., 2015).
The focus of the present study is the western Pacific and, in particular, the region of the South
China Sea. During NH winter the region is heavily influenced by the large anticyclone that
forms over Siberia each year which gives rise to strong north-easterly winds that impact deep
into the tropics as far south as Malaysia, Singapore and Indonesia. These north-easterly winds
typically prevail for 4-5 months (November-March) and form part of the East Asian winter
monsoon circulation. Superimposed on this seasonal synoptic flow are transient disturbances
known as cold surges, which are triggered by a southward shift of the anticyclone and lead to
sudden drops in surface air temperatures and increased wind speeds (Zhang et al., 1997;
Garreaud, 2001). It has been proposed that during these events significant amounts of
pollution from continental East Asia (>35°N) can be transported rapidly to the tropics (Ashfold
et al., 2015). Furthermore, these events, which can last for many days, occur regularly each
winter and are associated with some of the strongest convective activity, both in the western
Pacific region and globally. Indeed, trajectory calculations show that it can take less than 10
days for air masses to travel from the East Asian boundary layer (>35°N) to the upper tropical
troposphere (altitudes > 200 hPa), thereby providing a fast route by which VSLS (and many
other pollutants) may enter the lower stratosphere, despite their shorter lifetimes (Ashfold et
al., 2015).
Here we provide strong evidence to support this proposed transport mechanism based on new
atmospheric observations in the East and SE Asia region. We will present new Cl-VSLS
measurements from recent ground-based and aircraft campaigns in the region during which
we have observed dramatic enhancements in a number of Cl-VSLS, including $CH_2Cl_2$, 1,2-
dichloroethane ($CH_2ClCH_2Cl$), trichloromethane ($CHCl_3$) and tetrachloroethene ($C_2Cl_4$).
Furthermore we will demonstrate how pollution from China and the surrounding region can
rapidly, and regularly, be transported across the South China Sea and subsequently uplifted
to altitudes of 11-12 km, the region close to the lower TTL. Using the NAME particle dispersion
model we will also investigate the origin of the observed Cl-VSLS and examine the frequency
and duration of cold surge events. Finally we present some new estimates of $CH_2Cl_2$



emissions from East Asia and use these to estimate the likely emissions of $CH_2ClCH_2Cl$, for
which there is little information in the recent literature.
## 2. Methods
Between 2012 and 2014, air samples were collected at various times at (1) two coastal sites
in Taiwan, Hengchun (22.0547°N, 120.6995°E) and Fuguei Cape (25.297°N, 121.538°E); (2)
the Bachok Marine Research Station on the Northeast coast of Peninsular Malaysia (6.009°N,
102.425°E); and (3) during several flights of the CARIBIC aircraft between Germany and
Thailand/Malaysia (http://www.caribic-atmospheric.com/).
### 2.1 Sample collection
Air samples from Taiwan and Malaysia were collected in 3.2 litre silco-treated stainless steel
canisters (Restek) at a pressure of approximately 2 bar using a battery-powered diaphragm
pump (Air Dimensions, B series). In Taiwan the samples were collected from the surface via
a 1m x ¼" OD Dekabon sampling line, whilst in Bachok the samples were collected from the
top of an 18 m tower via a 5 m x ¼" OD Dekabon sampling line. In both cases the tubing was
flushed for at least 5 minutes prior to sampling. The sampling integrity was confirmed by
sampling high purity air (BTCA-178, BOC) through the inlet tubing and pump. Samples were
collected within 50 m of the sea and only when the prevailing winds were from the sea,
minimising the impact of any local emissions. The CARIBIC aircraft samples were collected in
2.7 litre glass flasks at a pressure of 4.5 bar using a two-stage metal bellows pumping system
(Brenninkmeijer et al., 2007; Baker et al., 2010) during flights between (i) Frankfurt (Germany)
and Bangkok (Thailand), and (ii) Bangkok and Kuala Lumpur (Malaysia). Samples were
generally collected at altitudes between 10 and 12 km.
### 2.2 Sample analysis
The collected air samples were shipped to UEA and analysed for their halocarbon content by
gas chromatography – mass spectrometry (GC-MS) following trace gas enrichment using
previously published methods. All samples (i.e. Taiwan, Bachok and CARIBIC) were analysed
for $CH_2Cl_2$, $CHCl_3$ and $C_2Cl_4$ using an Entech-Agilent GC-MS system operating in electron
ionisation (EI) mode, as described in Leedham-Elvidge et al., (2015). 1 litre samples were
dried and pre-concentrated before injection onto a 30 m x 0.32 mm GS Gas Pro capillary
column (Agilent), temperature ramped from -10°C to 200°C. Samples were interspersed with
repeated analyses of a working standard (SX-706070), a high pressure air sample contained
in a 34 litre electropolished stainless steel cylinder (Essex Industries) provided by the Earth
System Research Laboratory of the National Oceanic and Atmospheric Administration
(NOAA-ESRL, Boulder, CO, USA). $CH_2Cl_2$, $CHCl_3$ and $C_2Cl_4$ were quantified on ions with a
mass-to-charge ratio of 84 ($CH_2{}^{35}Cl_2{}^+$), 83 ($CH^{35}Cl_2{}^+$ and 166 ($C_2{}^{35}Cl_3{}^{37}Cl^+$) respectively. Mean
analytical precisions were ± 2% for $CH_2Cl_2$ and $C_2Cl_4$, and ± 3% for $CHCl_3$. Instrument blanks,
determined by analysing 1 litre aliquots of high purity nitrogen (BOC, Research grade), were
always below the detection limit of the instrument.
Some of the ground-based samples and a subset of the CARIBIC samples were also analysed
for a range of halocarbons, including the newly-identified $CH_2ClCH_2Cl$, using a pre-
concentration/GC system coupled to a Waters AutoSpec magnetic sector MS instrument, also
operating in EI mode, but run at a mass resolution of 1000 at 5 % peak height. Samples (using
between 200 and 250 ml of air) were analysed on an identical GS GasPro column following a



previously described method (Laube et al., 2010; Laube et al., 2012; Leedham-Elvidge et al.,
2015). $CH_2ClCH_2Cl$ was monitored on the ions with mass-to-charge ratios of 61.99 ($C_2H_3{}^{35}Cl^+$,
qualifier) and 63.99 ($C_2H_3{}^{37}Cl^+$, quantifier). Mean analytical precision was 1.4 % for
$CH_2ClCH_2Cl$ and the average blank signal was 0.07 ppt (as quantified using regular
measurements of research-grade helium) and was corrected for on a daily basis.
***2.3 Calibration and quality assurance***
$CH_2Cl_2$, $CHCl_3$ and $C_2Cl_4$ data are reported on the latest (2003) calibration scales provided by
NOAA-ESRL. As was shown in Leedham-Elvidge et al., (2015) our $CH_2Cl_2$ measurements
compare very well with those of NOAA-ESRL at our mutual long-term sampling site at Cape
Grim, Tasmania over more than 6 years. As a recognised international calibration scale for
$CH_2ClCH_2Cl$ is not yet available this compound was calibrated at UEA using the established
static dilution technique recently described (Laube et al., 2012). $CH_2ClCH_2Cl$ was obtained
from Sigma Aldrich with a stated purity of 99.8 %.Three dilutions were prepared at 7.1, 11.9
and 15.8 ppt. The mixing ratio assigned to our working standard from these dilutions was 5.67
ppt with a 1 σ standard deviation of 1.8 %. CFC-11 was added to the dilutions as an internal
reference compound and the CFC-11 mixing ratios assigned to the working standard through
these dilutions agreed with the value assigned by NOAA-ESRL within 4.3 %. This is well within
the estimated uncertainty of the calibration system of 7 % (Laube et al., 2012). In addition the
mixing ratios of $CH_2ClCH_2Cl$ in the working standard were compared with those in three other
high-pressure canisters (internal surface was either electropolished stainless steel or
passivated aluminium) over the whole measurement period. The ratios between standards did
not change within the 2 σ standard deviation of the measurements for any of the canisters
analysed indicating very good long-term stability for $CH_2ClCH_2Cl$. This was also the case for
$CHCl_3$ and $C_2Cl_4$. As noted in Leedham-Elvidge et al., (2015) mixing ratios of $CH_2Cl_2$ were
found to change over longer timescales in some of our standard canisters, but this drift has
been successfully quantified and corrected for as indicated by the very good comparability
with NOAA-ESRL measurements at the Cape Grim site noted above.
**3. Results**
Figure 1 shows the location of the three surface observation stations as well as the location of
the CARIBIC samples. The aircraft sampling points have been coloured by their $CH_2Cl_2$
concentration (see later discussion). Data from the surface stations and from the CARIBIC
aircraft flights are summarised in Table 1, together with a summary of published observations
as reported in the most recent Scientific Assessment of Stratospheric Ozone Depletion
(Carpenter and Reimann et al., 2015).
The highest concentrations of chlorinated VSLS were measured in samples collected in
Taiwan, suggesting that Taiwan is located relatively close to major emission regions. Figure 2
shows the 2014 data from Cape Fuguei. The Numerical Atmospheric-dispersion Modelling
Environment model (NAME, see supplementary material) can be used to infer the recent
transport history of this pollution. Our NAME analysis (Fig. 2 b-d) indicates that most of the
samples that contained high concentrations of Cl-VSLS had originated from regions to the
north of Taiwan, primarily the East Asian mainland. The median sum of chlorine from the 4
VSLS listed above ($\Sigma Cl_{VSLS}$) in 22 samples collected at Cape Fuguei in March/April 2014 was
756 ppt (range 232-2178 ppt). Similarly high concentrations and variation were seen in the 21
samples collected at Hengchun in March/April 2013 (see supplementary material). To put



these concentrations in a global context, the total organic chlorine derived from all known
source gases in the background troposphere (including CFCs, HCFCs, and longer-lived
chlorocarbons) is currently around 3300 ppt, with a typical Cl-VSLS contribution in the remote
marine boundary layer of approximately 3 % (Carpenter and Reimann et al., 2015). Of the
four VSLS measured, the two largest contributors to $\Sigma Cl_{VSLS}$ in Taiwan were $CH_2Cl_2$ (55-76 %)
and $CH_2ClCH_2Cl$ (14-30 %).

Figure 3 shows the Cl-VSLS data from 25 samples collected at Bachok, Malaysia during the
winter monsoon season in January/February 2014. During this phase of the monsoon the
prevailing winds are from the northeast and, as described earlier, are often impacted by
emissions further to the north, including from mainland China. As can be seen in Figure 3,
there was a 7 day period between 19 and 26 January when significantly enhanced
concentrations of Cl-VSLS were observed. During this period NAME back trajectories (Fig. 3
b-d) show air travelling from continental East Asia and across the South China Sea before
arriving at Bachok. These trajectories often pass over Taiwan and, in some instances, also
over parts of Indochina where additional emissions could have been picked up. As in the
Taiwan samples, $CH_2Cl_2$ is the largest contributor to $\Sigma Cl_{VSLS}$ (59-66 %), having a mean
concentration of 179.9 ± 71.9 ppt (range 94.0 – 354.9 ppt, 9 samples) during the 7-day period
of the pollution event. The mean concentration of $CH_2ClCH_2Cl$ was 64.4 ± 23.9 ppt (range
30.2 – 119.5 ppt), accounting for 19-23 % of $\Sigma Cl_{VSLS}$. These abundances are substantially
higher than those typically found in the marine boundary layer. For example, the range of
$\Sigma Cl_{VSLS}$ from the 4 compounds listed above in the tropical marine boundary layer reported in
WMO (2014) is 70-134 ppt. The range observed at Bachok over the entire sampling period
was 207-1078 ppt, with medians of 546 ppt and 243 ppt during the polluted (20-26 Jan) and
less-polluted (27 Jan – 6 Feb) periods respectively (see Table 1). It is interesting to note that
even in the period after the cold surge event (Fig.3 e,f), the levels of Cl-VSLS are still
significantly higher than would be expected, suggesting that this region of the South China
Sea is widely impacted by emissions from E Asia.

The pollution or "cold surge" event observed at Bachok lasted for 6-7 days and the back
trajectories shown in Figure 3 are typical of those arriving at Bachok during the winter
monsoon period (see NAME animations in supplement). To further investigate the frequency
and typical duration of these events a NAME trajectory analysis using carbon monoxide (CO)
as a tracer of industrial emissions from regions north of 20°N was conducted for the entire
winter season (see supplementary information for details). Figure 4(a) shows a time series of
this industrial CO tracer for winter 2013/2014 and suggests that the observed event in January,
during which there was a strong correlation between the industrial CO tracer and $CH_2Cl_2$ (Fig.
4b), is likely to be repeated regularly throughout the winter. An analysis of a further 5 winters
(Fig. 4c) demonstrates that 2013/14 was not unusual and that the events depicted in Figure
3a occur repeatedly every year (see supplement for further details).

The Bachok measurements clearly demonstrate the rapid long-range transport of highly
elevated concentrations of Cl-VSLS for several thousand kilometres across the South China
Sea, as predicted by Ashfold et al., (2015). However, to have an impact on stratospheric ozone
it is necessary to demonstrate that these high concentrations of Cl-VSLS can be rapidly lifted
to the upper tropical troposphere (lower TTL) or above. Such evidence can be found in
samples from several recent CARIBIC aircraft flights in the Southeast Asia region. Figure 1
shows significant enhancements of $CH_2Cl_2$ during flights over northern India and the Bay of



Bengal, and also between Bangkok and Kuala Lumpur. The same data is plotted against
longitude in Figure 5(a) which shows that elevated concentrations were observed in all
CARIBIC flights in the region (7) during the periods Dec 2012 - Mar 2013 and Nov 2013 - Jan
2014. The samples were collected in the altitude range 10-12 km, showing that recent
industrial emissions can regularly reach the lower boundary of the TTL. Although $CH_2ClCH_2Cl$
was only analysed for in a selection of samples during the flights from Germany to Bangkok,
elevated mixing ratios coinciding with the high levels of $CH_2Cl_2$ were clearly observed (Fig.
5b). $CHCl_3$ and $C_2Cl_4$ were also enhanced during these flights (Table 1), with $\Sigma Cl_{VSLS}$ being in
the range 48-330 ppt (Fig. 5c). This is up to 3.2 times higher than that previously found in the
lower TTL (36-103 ppt; Carpenter and Reimann et al., 2015). The highest abundances of Cl-
VSLS were seen in samples collected over the Bay of Bengal, and on flights between Bangkok
and Kuala Lumpur (Fig. 5a). NAME back trajectories (Fig. 5d) indicate that in these cases the
sampled air had almost always been transported from the east, and had often been impacted
by emissions from East Asia, with possible contributions from other countries including the
Philippines, Malaysia and Indochina.
**4. Discussion**
The high mixing ratios of $CH_2Cl_2$ observed in the Taiwan samples are not entirely unexpected.
Previous studies have found very high levels (> 1 ppb) of $CH_2Cl_2$ in various Chinese cities
(Barletta et al., 2006) and in the Pearl River Delta region (Shao et al., 2011). Elevated levels
(several hundred ppt) were also observed in aircraft measurements in polluted air emanating
from China during the TRACE-P campaign in 2001 (Barletta et al., 2006). These studies took
place in the early 2000s and emissions may be expected to have grown significantly since.
$CH_2Cl_2$ is predominantly (~90%) anthropogenic in origin, and is widely used as a chemical
solvent, a paint stripper and as a degreasing agent (McCulloch and Midgely, 1996; Montzka
et al., 2011). Other uses include foam blowing and agricultural fumigation. A growing use of
$CH_2CL_2$ is in the production of HFC-32 ($CH_2F_2$), an ozone friendly replacement for HCFC-22
($CHF_2Cl$) in refrigeration applications. Around 10% of global $CH_2Cl_2$ emissions come from
natural marine and biomass burning sources (Simmonds et al., 2006; Montzka and Reimann
et al., 2011).
Whilst the strong enhancements of $CH_2Cl_2$ are not entirely unexpected, the presence of high
concentrations of $CH_2ClCH_2Cl$ most certainly are. There are very few previously reported
measurements of $CH_2ClCH_2Cl$, particularly in recent years. Elevated levels have been
observed in urban environments close to known emission sources (Singh et al., 1981) and,
more recently, Xue et al., (2011) reported elevated levels (91 ± 79 ppt) in air samples collected
in the boundary layer over north-eastern China in 2007. The few reported measurements of
$CH_2ClCH_2Cl$ in the remote marine boundary layer are typically in the low ppt range (see Table
1) but these were made well over a decade ago. No long-term atmospheric measurements of
$CH_2ClCH_2Cl$ have been reported, and $CH_2ClCH_2Cl$ is not reported by the main surface
monitoring networks (AGAGE and NOAA), so current background concentrations and longer
term trends are unknown. $CH_2ClCH_2Cl$ is exclusively anthropogenic in origin, its primary use
being in the manufacture of vinyl chloride, the precursor to polyvinyl chloride (PVC), and a
number of chlorinated solvents. $CH_2ClCH_2Cl$ also finds use as a solvent, dispersant and has
historically been added to leaded petrol as a lead scavenger (EPA, 1984). In common with
$CH_2Cl_2$ it has also used as a cleaning/degreasing agent and as a fumigant. China is the world's
largest producer of PVC accounting for 27% of global production in 2009 (DCE, 2017).





Production has increased rapidly in recent years (14% per year over the period 2000-2009),
which could potentially have led to increased atmospheric emissions of $CH_2ClCH_2Cl$.
Based on an analysis of chlorocarbon production, sales and import/export figures, we have
estimated annual $CH_2Cl_2$ emissions from China to be of the order of 440-615 kilotonnes (kt)
$yr^{-1}$ (see supplementary material for methods). A simple correlation of $CH_2Cl_2$ and $CH_2ClCH_2Cl$
mixing ratios from the 2014 Bachok data ($R^2 = 0.9799$) would then imply Chinese $CH_2ClCH_2Cl$
emissions of 163-227 kt $yr^{-1}$. If true, the scale of these emissions is a major surprise as
$CH_2ClCH_2Cl$ is highly toxic and believed to be used almost exclusively in non-emissive
applications.
The other Cl-VSLS presented here are $C_2Cl_4$ and $CHCl_3$. In contrast to $CH_2ClCH_2Cl$, long-term
atmospheric data records are available for these compounds, although there are few data
from the SE Asia region. Current trends show that $C_2Cl_4$ is declining in the background
troposphere (~6 % $yr^{-1}$), whilst $CHCl_3$ is approximately constant (Carpenter and Reimann et
al., 2015). However, both compounds were elevated in the samples containing high
concentrations of $CH_2Cl_2$ and $CH_2ClCH_2Cl$, suggesting that significant, co-located sources
remain. Like $CH_2ClCH_2Cl$, $C_2Cl_4$ is exclusively anthropogenic in origin, used primarily as a
solvent in the dry cleaning industry, as a metal degreasing agent and as a chemical
intermediate, for example in the manufacture of the hydrofluorocarbons HFC-134a and HFC-
125. $CHCl_3$ is believed to be largely natural in origin (seawater, soils, macroalgae), but
potential anthropogenic sources include the pulp and paper industry, water treatment facilities
and HFC production (McCulloch, 2003; Worton et al., 2006; Montzka et al., 2011).
When calculating the VSLS contribution to stratospheric chlorine, it is usual to assume an
average concentration in the region of the TTL known as the level of zero radiative heating
(LZRH). The LZRH is located at the transition between clear-sky radiative cooling and clear-
sky radiative heating. This occurs at an approximate altitude of 15 km and it is believed that
air masses above this level will go on to enter the stratosphere[1]. As noted above there are
very few measurements in this region and, furthermore, many of the available measurements
were made over a decade ago and assumptions based on surface temporal trends have to be
made in order to estimate present day values (Carpenter and Reimann et al., 2015: Hossaini
et al., 2015). Another key deficiency in this estimation of VSLS concentrations entering the
stratosphere is that most of the reported measurements have not been made in the 2 key
regions where the strongest troposphere to stratosphere transport occurs. Although we have
no data from the region of the LZRH, the CARIBIC data over northern India and SE Asia
suggests that the contribution of VSLS to stratospheric chlorine loading may be significantly
higher than is currently estimated (50-95 ppt, Carpenter and Reimann et al., 2015). It is also
interesting to note that the much-discussed contribution of VSLS-Br compounds to
stratospheric bromine is approximately 5 ppt, which is equivalent to 300 ppt of chlorine (1 ppt
of bromine is roughly equivalent to 60 ppt chlorine, Sinnhuber et al., 2009). The CARIBIC
measurements suggest that Cl-VSLS could currently, on occasion, contribute a similar
amount.
These new measurements of Cl-VSLS in Taiwan, Malaysia and from an aircraft flying above
South-East Asia show that there are substantial regional emissions of these compounds; that
these emissions can be rapidly transported long distances into the deep tropics; and that an
equally rapid vertical transport to the upper tropical troposphere is a regular occurrence.



Although the focus of this paper is short-lived chlorinated gases, there are many other
chemical pollutants contained in these air masses which will have a large impact on regional
air quality, etc.
Unlike the bromine-containing VSLS which are largely natural in origin, the Cl-VSLS reported
here are mainly anthropogenic and consequently it would be possible to control their
production and/or release to the atmosphere. Of particular concern are the rapidly growing
emissions of $CH_2Cl_2$, and potentially $CH_2ClCH_2Cl$, especially when considering the
geographical location of these emissions, close to the major uplift regions of the western
Pacific (winter) and the Indian sub-continent (summer). Without a change in industrial
practices the contribution of Cl-VSLS to stratospheric chlorine loading is likely to increase
substantially in the coming years, thereby endangering some of the hard-won gains achieved,
and anticipated, under the Montreal Protocol.

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





**Acknowledgements**

The authors would like to thank our CARIBIC project partners and the CARIBIC technical team (in particular C. Koeppel, D. Scharffe). The Malaysia and Taiwan activities were funded through the UK Natural Environment Research Council (NERC) International Opportunities Fund (NE/J016012/1, NE/J016047/1, NE/N006836/1). CARIBIC has become part of IAGOS (www.IAGOS.org) and is supported by the German Ministry of Education and Science and Lufthansa. The CARIBIC halocarbon measurements were part-funded by the European FP7 project SHIVA (226224). UM-BMRS is supported by the Malaysian Ministry of Higher Education (Grant MOHE-HICoE IOES-2014). The sampling at Hengchun and Fugei Cape was operated under the Seven South East Asian Studies (7-SEAS) program and funded by Taiwan EPA and MOST. J.L and L.G. were funded through a NERC fellowship (NE/1021918/1) and studentship (NE/1210143) respectively. We acknowledge use of the NAME atmospheric dispersion model and associated NWP meteorological data sets made available to us by the UK Met Office. We also acknowledge the significant storage resources and analysis facilities made available to us on JASMIN by STFC CEDA along with the corresponding support teams.






**Table 1:** Summary of VSLS-Cl data from the 3 surface stations and the 7 CARIBIC flights. For
comparison, the ranges reported in the most recent WMO Ozone Assessment (Carpenter and
Reimann et al., 2015) for the marine boundary layer (MBL) and lower Tropical Tropopause
Layer (TTL, 12-14 km altitude) are also shown.

**Figure 1:** Map of the region showing the location of each CARIBIC sample. The markers have
been coloured according to their $CH_2Cl_2$ concentration to highlight the regions where
enhanced levels of VSLS were observed. Also shown are the approximate locations of the 3
surface stations

**Figure 2:**
*Upper panel (a):* Mole fractions (ppt) of the 4 chlorinated VSLS in air samples collected at
Cape Fuguei, Taiwan in March/April 2014. The error bars are ± 1 standard deviation. The
black arrows show the dates of the footprint maps shown below.
*Lower panel (b-d)*: NAME footprint maps indicating the likely origin of the air sampled at Cape
Fuguei. Figures (b, 13 March) and (c, 30 March) show examples where the observed VSLS
levels are very high and suggest a strong influence from continental East Asia. Figure (d) is
from 29 March where the influence of the mainland is much lower and the VSLS mole fractions
are much closer to the expected background level.
The location of Cape Fuguei is indicated with a blue circle (see also Figure 4)

**Figure 3:**
*Upper panel (a)*: Mole fractions (ppt) of the 4 chlorinated VSLS in air samples collected at
Bachok in Jan/Feb 2014. Strongly enhanced levels of all 4 compounds were seen for a 7-
day period at the beginning of the campaign (20-26 Jan). Also shown (dashed line) are the
reported median background concentrations in the remote marine boundary layer in 2012[1].
*Lower panels (b-f)*: NAME footprint maps indicating the likely origin of the air sampled at
Bachok. During the pollution episode (b = 21 Jan; c = 23 Jan; d = 24 Jan) the samples would
have been heavily impacted by emissions from the East Asian mainland, whilst this influence
is much reduced during the cleaner, non-polluted periods (e = 3 Feb; f = 5 Feb). Note that
even after the main pollution event, the abundance of the VSLS remain significantly above
true background levels for much of the time, suggesting a widespread influence from
industrial emissions on a regional scale.
The location of Bachok is indicated with a blue circle (see also Figure 4)
**Figure 4:**
(a) Time-series of the modelled carbon monoxide (CO) anomaly at Bachok, due only to
industrial emissions from north of 20°N in the previous 12 days, for winter 2013/14. The
$CH_2Cl_2$ data (grey squares) from the Bachok sampling period are overlaid. The dashed lines
show the 25 ppb and 50 ppb thresholds referred to in 3c (see supplement for further details).
(b) Correlation of the modelled CO anomaly with the observed $CH_2Cl_2$.
(c) Average number of days each month, averaged over six consecutive winters (2009/10 –
2014/15) where the modelled carbon monoxide anomaly at Bachok is above a particular
threshold (25 ppb and 50 ppb which, from the regression in 3b, correspond to 176 ppt and



315 ppt of CH$_2$Cl$_2$). The 2013/14 winter is shown separately for comparison with the 6-year
average.
**Figure 5**
(a) Mole fractions (ppt) of CH$_2$Cl$_2$ in CARIBIC air samples collected at 10-12km altitude over
Northern India, the Bay of Bengal and SE Asia. The samples are plotted against longitude
and have been coloured by date.
(b) Mole fraction (ppt) of CH$_2$ClCH$_2$Cl in selected CARIBIC samples (note: CH$_2$ClCH$_2$Cl was
not monitored in the samples collected between Bangkok to Kuala Lumpur, and only in a
selection of samples on the Frankfurt-Bangkok route).
(c) Total VSLS-Cl derived from the 4 compounds of interest in the CARIBIC samples (note:
total Cl-VSLS could only be calculated for the samples shown in Fig 5b above).
(d) NAME footprint maps indicating the likely origin of the air sampled by the CARIBIC
aircraft. NAME footprints at this altitude, and particularly in regions of strong sub-grid-scale
convection not captured fully in the gridded meteorological input data, may be less reliable
than those at the surface sites. This makes pinpointing particular emission regions more
difficult. The central panel therefore shows a composite footprint derived from the samples
that contained the highest levels of CH$_2$Cl$_2$ (90[th] percentile, [CH$_2$Cl$_2$] >75.6 ppt), with the
composite footprint from the remaining samples ([CH$_2$Cl$_2$] < 75.6 ppt) shown in the left hand
panel. To emphasise the likely source regions the right hand panel shows the difference
between the middle and left hand panels.  The geographical location of each sample
included in the composite analysis are shown in blue circles.





**Table 1**

| | Taiwan 2013 | | Taiwan 2014 | | | Bachok 2014 | | MBL (WMO 2014) [b] | |
|---|---|---|---|---|---|---|---|---|---|
| | *Median* | *Range* | *Median* | *Range* | *Median (CS)* [a] | *Median (non-CS)* | *Range* | *Median* | *Range* |
| CH$_2$Cl$_2$ | 226.6 | 68 - 624 | 227.4 | 70 - 639 | 170.4 | 81.9 | 64.8 – 355 | **28.4** | **21.8 – 34.4** |
| CH$_2$ClCH$_2$Cl | - | - | 85.4 | 16.7 – 309 | 62.2 | 21.7 | 16.4 – 120 [c] | **3.7** | **0.7 - 14.5** [d] |
| CHCl$_3$ | 33.0 | 11.6 – 232 | 35.1 | 13.8 - 103 | 22.8 | 14.7 | 12.8 – 30.5 | **7.5** | **7.3 – 7.8** |
| C$_2$Cl$_4$ | 4.4 | 1.7 – 16.6 | 5.5 | 1.7 – 18.6 | 4.5 | 1.9 | 1.5 – 9.5 | **1.3** | **0.8 – 1.7** |
| Σ Cl$_{VSLS}$ | - | - | 755.8 | 232 -2178 | 546.0 | 243.1 | 207 – 1078 [c] | **93.4** | **70 - 134** |

| | CARIBIC (FFT-BKK, 65-97°E) | | | CARIBIC (BKK-KUL, 100-105°E) | | | Lower TTL (WMO 2014) [b] | |
|---|---|---|---|---|---|---|---|---|
| | 10-12 km | | | 10-12 km | | | 12-14 km | |
| | *Mean* | *Median* | *Range* | *Mean* | *Median* | *Range* | *Mean* | *Range* |
| CH$_2$Cl$_2$ | 43.2 | 31.6 | 14.6 - 121 | 50.4 | 46.5 | 22.5 - 100 | **17.1** | **7.8 – 38.1** |
| CH$_2$ClCH$_2$Cl [e] | 9.9 | 6.1 | 0.4 – 29.1 | - | - | - | **3.6** | **0.8 – 7.0** |
| CHCl$_3$ | 7.0 | 6.0 | 2.0 – 15.6 | 9.3 | 8.7 | 3.7 - 46.6 | **6.8** | **5.3 – 8.2** |
| C$_2$Cl$_4$ | 0.87 | 0.65 | 0.1 – 4.4 | 1.6 | 1.5 | 0.2 - 5.9 | **1.1** | **0.7 – 1.3** |
| Σ Cl$_{VSLS}$ [e] | 153.7 | 119.3 | 48.4 - 330 | | | | **67** | **36 - 103** |
| *Σ Cl$_{VSLS}$* * [f] | - | - | - | *134.8* | *127.8* | *56.6 – 251* | - | - |

[a] CS and non-CS refer to the cold surge (polluted) and non-cold surge periods at Bachok

[b] The WMO data is a compilation of all reported global measurements up to, and including, the year 2012. The range represents the smallest mean minus one standard deviation and the largest mean plus one standard deviation of all considered datasets.

[c] CH$_2$ClCH$_2$Cl was only analysed for in 16 of the 28 samples collected at Bachok

[d] Note that the CH$_2$ClCH$_2$Cl MBL data actually date back to the early 2000s. No recent data was reported.

[e] CH$_2$ClCH$_2$Cl was only analysed for in selected samples from the Frankfurt-Bangkok flights and in no samples collected during the Bangkok-Kuala Lumpur flights

[f] Σ Cl$_{VSLS}$* is defined as the sum of VSLS-Cl excluding the contribution from CH$_2$ClCH$_2$Cl.





**Figure 1**



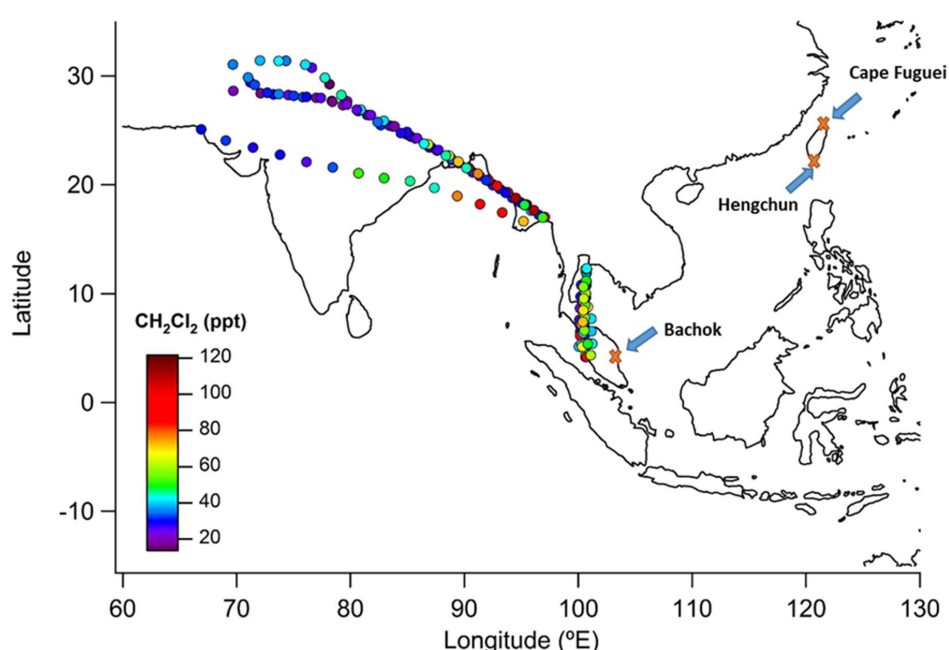










**Figure 2**

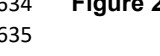

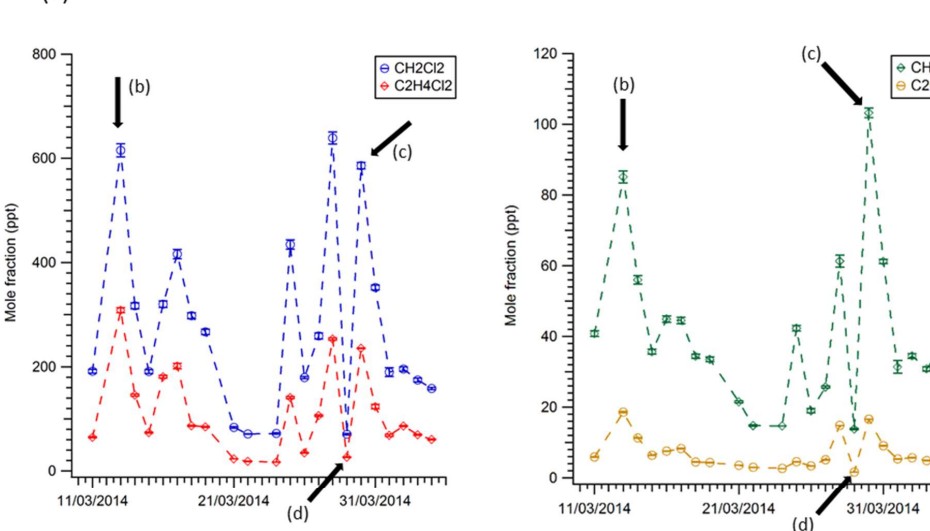

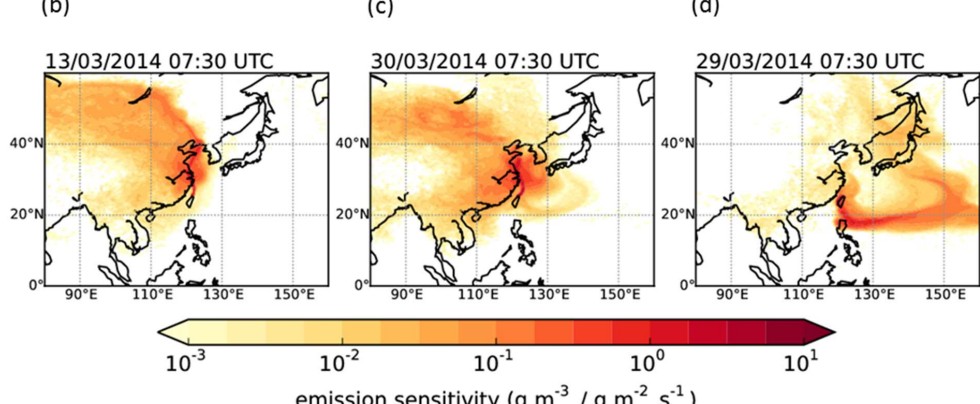




**Figure 3**

Figure 3





**Figure 4**

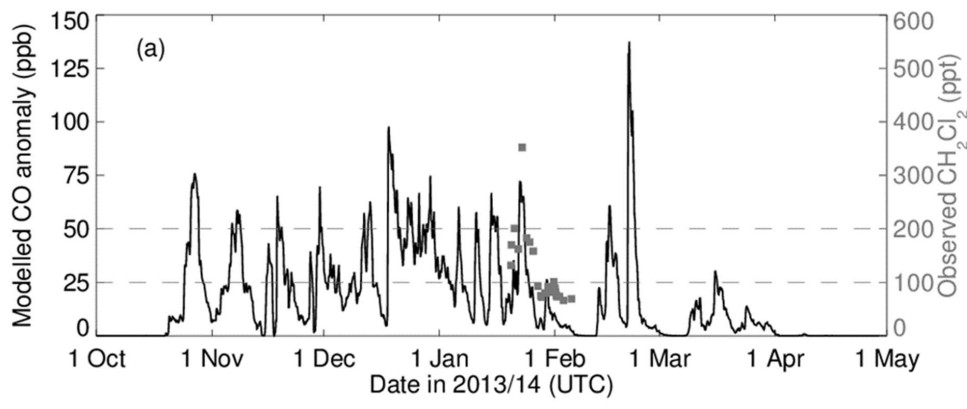

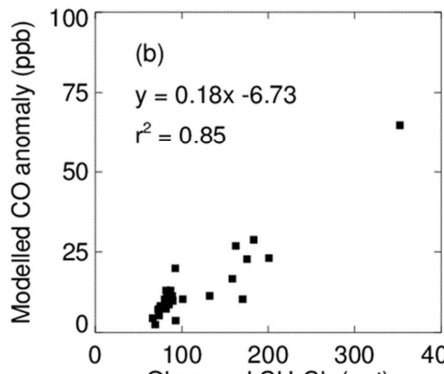
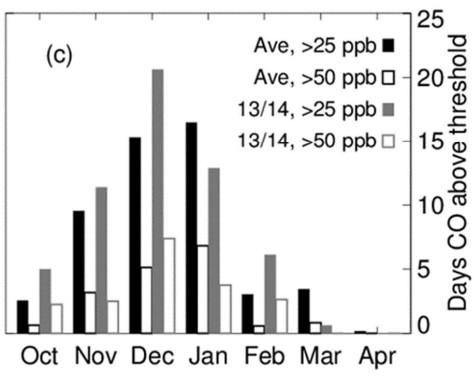



**Figure 5**

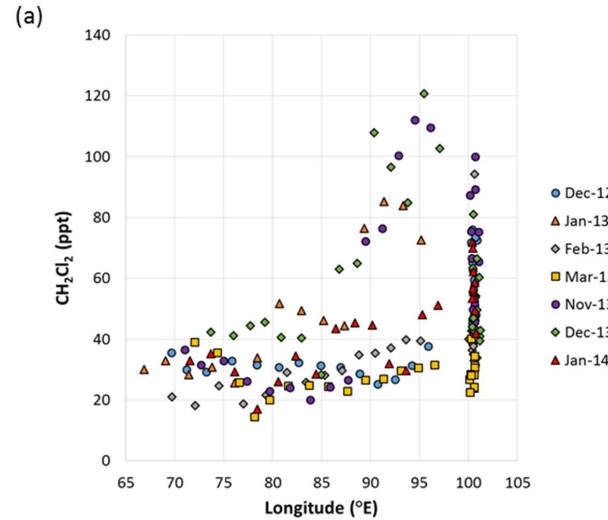

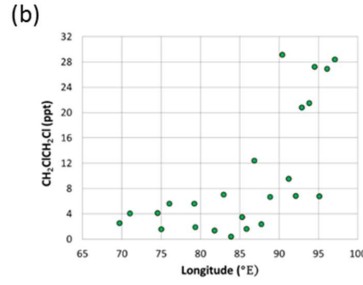

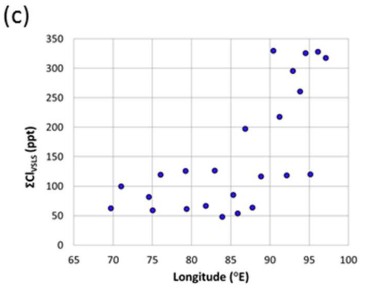

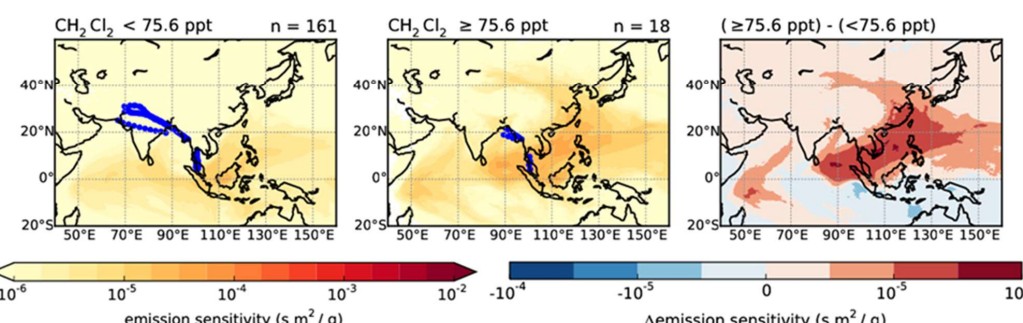
