# Peer review of "A growing threat to the ozone layer from short-lived anthropogenic chlorocarbons"

_Atmospheric Chemistry and Physics, 2017_

## Referee Comment (RC1) · B.-M. Sinnhuber (Referee) · 11 Jul 2017

The manuscript by Oram et al. reports on observations of short-lived chlorocarbons in East and South East Asia from surface measurements, as well as aircraft measurements in the upper troposphere. These observations show much larger values than previously reported in the tropical marine boundary layer or upper troposphere. Recent studies already have found a large increase in $CH_2Cl_2$. In the present study, new estimates of $CH_2Cl_2$ emissions from East Asia are presented and used to present first estimates of the emission of $CH_2ClCH_2Cl$. This study seems well performed, it is highly relevant and timely and the manuscript is well written. I recommend publication in Atmos. Chem. Phys. after consideration of only a few minor comments.

[Figure]

As one general comment I feel that the presentation of CH2Cl2 emission estimates and the correlation with CH2ClCH2Cl to infer new CH2ClCH2Cl emission estimates (lines 337 - 343) deserves (and requires) more detail, given its importance. Part of the information on estimating the CH2Cl2 emissions given in the supplementary material should be included in the main text and a bit more explanation on the "simple correlation" should be included.

Specific comments and corrections

Abstract, l.31: "higher than expected": what is this expectation based upon? Based on previously reported measurements? On line 360 and following it is discussed that many of the previous measurements have been made over a decade ago and in different regions ("...not the 2 key regions..."). I believe it would be good to make a bit clearer from the start if the enhancements seen in this study are likely because of recent increases in emissions, regional differences, or both.

l.36: define "Cl-VSLS" when first used. Moreover better use this consistently throughout (e.g. Table 1 uses VSLS-CL, which I suppose means the same)

l.45: you may want to cite also the recent study by Hossaini et al., The increasing threat to stratospheric ozone from dichloromethane, Nature communications, 2017, that was published after submission of the present manuscript.

l.104: I suggest to break the sentence in two: "...in the TTL. Surface measurements ..."

l. 126: I don't understand the meaning of "globally" here.

l.129: "shorter lifetimes" could be misleading here, as it may imply lifetimes shorter than the 10 days for air masses to travel from East Asia to the TTL, which is probably not what is meant?

l.150: "the CARIBIC aircraft": better include a sentence or two on the CARIBIC project, describing that these are measurements from in-service aircrafts, ideally including a

reference paper (in addition to the http link).

l.328: "CH2ClCH2Cl is exclusively anthropogenic in origin...": WMO (2014) lists also biomass burning as a source of CH2ClCH2Cl. Can you include references on additional sources?

l.334: "Production has increased rapidly...": Can you give a reference for this increase in production?

l. 362: Does the superscript "1" have any meaning? Footnote?

l. 367: "2"-> "two"

Table 1: Why not use the IATA code "FRA" for Frankfurt (rather than "FFT", which is the IATA code for Frankfort, Kentucky)?

Table 1: Why is the sum of VSLS-CL excluding CH2ClCH2Cl not given for the other data for comparison?

---

## Referee Comment (RC2) · Anonymous Referee #2 · 22 Jul 2017

The manuscript titled "A growing threat to the ozone layer from short-lived anthropogenic chlorocarbons" present a very interesting and useful study of very short-lived substances (VSLS), particularly Chlorine (Cl). These substances are not regulated under the Montreal Protocol, although they would contribute to the ozone depletion. The authors have used measurement from IAGOS-CARIBIC and surface stations in East and South East Asia. The observations show higher values than expected or reported in the most recent Scientific Assessment of Stratospheric Ozone Depletion (Carpenter and Reimann et al., 2015). This study shows further evidence of rapid transport of CI-VSLS toward the tropics and subsequently uplifted to the lower Tropical Tropopause Layer (lower TTL, 10-12km) using a lagrangian particle dispersion (NAME) and high quality in situ data (IAGOS-CARIBIC).

The manuscript is well written and the quality of the text is very much appreciated. I suggest minor revisions. You can find below general comments followed by more specific comments.

General comments:

Methods/Sample collection section:

In general, all the information on the samples: time, locations for all the data, number of flights for CARIBIC data, number of sampling at the ground-based stations should be added in this section. For instance, the 7 IAGOS-CARIBIC flights time should be mentioned. Days and months of samples should be specified in this section as well. It will help the reader to get the general feature of the sampling.

The altitude of CARIBIC needs to be shown. Have you filtered IAGOS-CARIBIC data to analyze data between 10 and 12 km only? The statistics of the sampling in this layer is needed.

In the text it is mentioned that 10-12km over East Asia is the lower boundary of the TTL. It would be very helpful to show a map of TTL or a figure of TTL and aircraft altitudes together with respect of the flight tracks (latitude). It would be also useful to directly refer to Box 1-3, Figure 1 of Carpenter and Reimann et al. (2015) that shows the altitude range of the TTL.

Results section:

About the results shown in Figure 3, a sentence explaining that three days have been chosen out of the seven days of the cold surge event would be helpful. The term "cold surge" should be mentioned.

In general, "see in supplement" is largely used in the manuscript but I would suggest to refer to figure number and section names of the supplement materials to help the reader. Results from Carpenter and Reimann et al., 2015 are cited as reference for Interactive comment

CI-VSLS last information. For results at the lower TTL, it would be useful to recall the type of observations used in the assessment report: aircraft campaigns and balloons.

Specific comments:

Table: Units need to be added in Table 1 and its caption

**Figures:**

In general, the way to write CI-VSLS should be consistent along the captions and the text (sometimes chlorinated VSLS or VSLS-CI).

Figure 1: "(red crosses)" for surface sites on the map need to be added in the caption to guide the reader.

Figure 2: We don't see the blue circle on the map.

Figures 2,3,5: Helpful to have a recall of ground-based stations location.

Figure 2 and 3: I would rather use letter to name the panels in the caption and I would rather use numbers to link plots in upper panel with maps of the bottom panels.

Figure 3: Use arrows as for Figure 2 to help the reader to find the days that are chosen for air masses origin (map below). In the caption it is mentioned "true background levels", how these levels are estimated? Figure 3 a) is not specified, "a)" should be added on the figure.

Figure 4 and S2: What does CO anomaly mean? What is the reference value?

Text:

Line 126: "Both in the western Pacific region and globally". It is not clear what globally means.

Line 147: "Various time" needs to be specified (see general comments).

Line 150: Change "CARIBIC aircraft" to "IAGOS-CARIBIC aircraft" as CARIBIC is part of the IAGOS program.

Line 230: Change "shows the 2014 data..." to "shows data from CAPE Fuguei in the
end of March, beginning of April 2014".

Line 237: Change "March/April 2013" to "mid of March/beginning of April 2013".

Line 246: "January/February": the entire months are not shown so "end of January/beginning of February" would be more appropriate. "During this phase of the monsoon": A recall about the Asian Monsoon phases and references would be useful, maybe recall that it is the East Asian winter monsoon circulation as mentioned in the introduction.

Line 252: change "often" to "most of the case studies we are analyzing here" or "for N days out of Ntotal days of observations" or give the information in Line 277: Change "(see supplement for further details)" to "(Fig. S2 in the supplement material)".

Line 287: Change "in all CARIBIC flights" to "in the seven CARIBIC flights" and remove "(7)".

Line 334: Need a reference.

Line 362: Need a reference.

Line 383: Examples of other chemical pollutants would be useful.

Line 384: Remove "etc".

---

## Author Response (AR1)

[revised manuscript text omitted]

**Supplementary material**

**1. Additional measurements**

As mentioned in the main document, 21 air samples were also collected at the Hengchun station in southern Taiwan (22.0547N, 120.6995E) during March and early April 2013 (Figure S1). Unfortunately $CH_2ClCH_2Cl$ was not analysed for in the 2013 samples, but the absolute concentrations and variability of $CH_2Cl_2$, $CHCl_3$ and $C_2Cl_4$ were very similar to those observed at Cape Fuguei in 2014 (see also Table 1).

**2. Modelling**

The Numerical Atmospheric-dispersion Modelling Environment (NAME, Jones et al. 2007) is a Lagrangian particle dispersion model, used here to understand the origin of the sampled air masses. For each air sample, NAME was used to calculate batches of 60000 inert backward trajectories. For the ground samples the trajectories started at the measurement site within an altitude range of 0-100 m and were started throughout a 3 hour period encompassing the sample time. For the aircraft samples the trajectories were started at the exact time, horizontal coordinates and altitude at which the sample was collected. The surface sample trajectories ran for 12 days and the aircraft sample trajectories for 20 days. Every 15 minutes the location of all trajectories within the lowest 100 m of the model atmosphere was recorded on a grid with a horizontal resolution of 0.5625° longitude by 0.375° latitude. From this information, and assuming a uniform surface air density consistent with a pressure of 1000 hPa and a temperature of 25°C, the sensitivity of the sampled air mass to surface emissions occurring in the previous 12 or 20 days within a particular grid cell can be derived (units $sm^2g^{-1}$).

The trajectories were calculated using three-dimensional meteorological fields produced by the UK Meteorological Office's Numerical Weather Prediction tool, the Unified Model (UM). These fields have a horizontal grid resolution of 0.35° longitude by 0.23° latitude and 59 vertical levels below ~30 km, and are available at 3 h intervals. Vertical velocities are obtained from the UM and available at grid nodes. The sub-grid-scale process of turbulence is parameterised in NAME (Morrison and Webster 2005). Another sub-grid scale process, convection, is not parameterised in our NAME calculations. However, past work (Heyes et al. 2009, Ashfold et al. 2012, Navarro et al. 2015) has shown atmospheric composition in the tropics can be interpreted using trajectories calculated with wind fields that, while not resolving individual up- and down-draughts, are consistent with large-scale convective activity.

**2.1 Multi-year NAME calculations**

By combining the emission sensitivities derived from NAME with a distribution of emissions it is possible to calculate a modelled mixing ratio of the emitted species, due only to emissions occurring within the timescale of the backward trajectories, at the measurement site (dimensionally, $sm^2g^{-1}$ x $gm^{-2}s^{-1}$ = dimensionless mixing ratio). We have used an inventory of industrial and combustion carbon monoxide (CO) emissions (RCP8.5 for 2005; Granier et al. 2011, Riahi et al. 2011), which are likely to be similarly distributed to VSLS-Cl (e.g. regression in Figure 4b and Shao et al. 2011), to model anomalous CO volume mixing ratios (i.e. those due only to these industrial emissions north of 20ºN) at Bachok at 3-hourly resolution for six recent NH winters (Oct-Apr 2009/10-2014/15). Figure 4a in the main paper shows this modelled quantity over winter 2013/14, during which the Bachok observations were made, as an example. The observed peak in VSLS-Cl is 1) captured well by the model, and 2) likely to be a regularly repeated event. In total during this winter there are ~57 days (i.e. 453 3-hour periods) with a modelled mixing ratio above a threshold of 25 ppb, and 19 days above 50 ppb. To demonstrate that winter 2013/14 was not unusual, the modelled CO anomalies for the other 5 winters examined are shown in Figure S2.

**2.2 NAME animations**

The two animations (Jan2014.mp4 and Feb2014.mp4) show 3-hourly NAME footprints of air arriving at Bachok in January and February 2014 and indicate where surface emissions have an influence on the composition of air arriving at the site. The animations give an indication of the frequency that air arriving at Bachok has been influenced by emissions from East Asia.

**Supplementary References**

Ashfold, M.J.; Harris, N.R.P.; Atlas, E.L.; Manning, A.J. & Pyle, J.A. Transport of short-lived species into the Tropical Tropopause Layer Atmospheric Chemistry And Physics, 12, 6309-6322, 2012.

Granier, C. et al., Evolution of anthropogenic and biomass burning emissions of air pollutants at global and regional scales during the 1980--2010 period, *Climatic Change*, **109**, 163-190, 2011.

Heyes, W. J.; Vaughan, G.; Allen, G.; Volz-Thomas, A.; Pätz, H.-W. & Busen, R. Composition of the TTL over Darwin: local mixing or long-range transport? Atmospheric Chemistry and Physics, 9, 7725-7736, 2009.

Jones, A., Thomson, D., Hort, M., and Devenish, B.: The U.K. Met Office's Next-Generation Atmospheric Dispersion Model, NAME III, in: *Air Pollution Modeling and Its Application XVII*, edited by Borrego, C. and Norman, A.-L., 580–589, Springer US, doi:10.1007/978-0-387-68854-1_62, 2007.

Morrison, N.L. and Webster, H.N.: An Assessment of Turbulence Profiles in Rural and Urban Environments Using Local Measurements and Numerical Weather Prediction Results, *Bound.-Lay. Meteorol.*, **115**, 223–239, doi:10.1007/s10546-004-4422-8, 2005.

Navarro, M.A.; Atlas, E.L.; Saiz-Lopez, A.; Rodriguez-Lloveras, X.; Kinnison, D.E.; Lamarque, J.-F.; Tilmes, S.; Filus, M.; Harris, N.R.P.; Meneguz, E.; Ashfold, M.J.; Manning, A.J.; Cuevas, C.A.; Schauffler, S.M. & Donets, V. Airborne measurements of organic bromine compounds in the Pacific tropical tropopause layer Proceedings of the National Academy of Sciences 112, 13789-13793, 2015.

Riahi, K. et al., RCP 8.5—A scenario of comparatively high greenhouse gas emissions. *Climatic Change*, **109**, 33-57, 2011.

Shao, M., Huang, D., Gu, D., Lu, S., Chang, C. and Wang, J. Estimate of anthropogenic halocarbon emission based on measured ratio relative to CO in the Pearl River Delta region, China. *Atmos. Chem. Phys.*, **11**, 5011-5025, 2011.

[Figure]

**Figure S1**: Mole fractions (ppt) of 3 chlorinated VSLS in air samples collected at Hengchun,
Taiwan in March/April 2013. Note that $CH_2ClCH_2Cl$ was not monitored in the 2013 samples.
The error bars are ± 1 standard deviation.

**Modelled mid-latitude industrial CO anomaly (ppb) at Bachok**

**Figure S2**: Time-series of the modelled carbon monoxide (CO) anomaly at Bachok, due only to industrial emissions from north of 20°N in the previous 12 days, for six winter seasons. The period of observations at Bachok during Jan and Feb 2014 is shaded in grey. Also shown are the number of days in each winter which exceed the 25 ppb and 50 ppb thresholds which, using the regression equation in Figure 4b, correspond to 176 ppt and 315 ppt of $CH_2Cl_2$.

                                    **Response to Reviewers**

We would like to thank the reviewers for their very helpful comments and we have
addressed these as follows (reviewer's comment in bold):

**Reviewer 1** (Sinnhuber)

**As one general comment I feel that the presentation of CH2Cl2 emission estimates**
**and the correlation with CH2ClCH2Cl to infer new CH2ClCH2Cl emission estimates**
**(lines 337 - 343) deserves (and requires) more detail, given its importance. Part of the**
**information on estimating the CH2Cl2 emissions given in the supplementary material**
**should be included in the main text and a bit more explanation on the "simple**
**correlation" should be included.**

We have moved the discussion about the emission estimates from the supplement into the
manuscript as suggested. We have also made a slight modification to our analysis and
rather than giving the extreme range of potential $CH_2Cl_2$ emissions as before (based on the
40:60 and 50:50 production ratios) we have opted to give an estimate based on the more
likely ratio of 45:55. This leads us to production and emission figures of 715 kt and 455 kt
respectively, with an approximate uncertainty of ± 10%.

We have also expanded the section on the correlation between the two compounds to
include the following text

*"There is a strong linear correlation between the observed $CH_2Cl_2$ and $CH_2ClCH_2Cl$ data at*
*both Bachok ($R^2$ = 0.9799) and Cape Fuguei ($R^2$ = 0.9189). Combining the datasets yields a*
*slope of 0.4456 ± 0.0194 ($R^2$ =0.9228). Using the emissions for $CH_2Cl_2$ derived above (455*
*kt) and making the assumptions that (1) all emissions originate in China and (2) there are no*
*significant relative losses of the two compounds since emission (lifetimes are 144 days for*
*$CH_2Cl_2$ and 65 days for $CH_2ClCH_2Cl$) we can estimate Chinese emissions of $CH_2ClCH_2Cl$ to*
*be of the order of 203 ± 9 kt yr$^{-1}$. If accurate, the scale of these emissions is a major surprise*
*as $CH_2ClCH_2Cl$ is highly toxic (suggesting that local emissions would be minimised) and*
*believed to be used almost exclusively in non-emissive applications."*

**Abstract, l.31: "higher than expected": what is this expectation based upon? Based**
**on previously reported measurements? On line 360 and following it is discussed that**
**many of the previous measurements have been made over a decade ago and in**
**different regions ("… not the 2 key regions …"). I believe it would be good to make a**
**bit clearer from the start if the enhancements seen in this study are likely because of**
**recent increases in emissions, regional differences, or both.**

By "higher than expected" we do indeed mean higher than previously reported data. This
was stated earlier in the sentence and refers to both our surface and aircraft measurements.
We compare our data with the most recent review (WMO, 2015) as we describe in the text
(Section 3, line 236-238) and in Table 1. The enhancements we observe are likely to result
from a combination of increasing emissions and the location of the measurements, although
based on our measurements alone, which are over a relatively short period of time, we have
no evidence that emissions are increasing. We do know from previous work that
atmospheric levels of $CH_2Cl_2$ have increased, which implies increasing emissions of this
compound. The long term trend of $CH_2ClCH_2Cl$ is unknown.

**l.36: define "Cl-VSLS" when first used. Moreover better use this consistently**
**throughout (e.g. Table 1 uses VSLS-CL, which I suppose means the same)**

Done

**I.45: you may want to cite also the recent study by Hossaini et al., The increasing threat to stratospheric ozone from dichloromethane, Nature communications, 2017, that was published after submission of the present manuscript.**

As our paper was published in ACPD before the Hossaini et al paper it does not seem appropriate to include it in our reference list at this stage.

**I.104: I suggest to break the sentence in two: "… in the TTL. Surface measurements …"**

Done

**l. 126: I don't understand the meaning of "globally" here.**

We have removed the word "globally".

**I.129: "shorter lifetimes" could be misleading here, as it may imply lifetimes shorter than the 10 days for air masses to travel from East Asia to the TTL, which is probably not what is meant?**

Text changed to "*despite their relatively short atmospheric lifetimes*"

**I.150: "the CARIBIC aircraft": better include a sentence or two on the CARIBIC project, describing that these are measurements from in-service aircrafts, ideally including a reference paper (in addition to the http link).**

Done

**I.328: "CH2ClCH2Cl is exclusively anthropogenic in origin …": WMO (2014) lists also biomass burning as a source of CH2ClCH2Cl. Can you include references on additional sources?**

We have added the following text: "*Simpson et al. (2011) observed a small enhancement in CH$_2$ClCH$_2$Cl in Canadian boreal forest fire plumes (background average, June-July 2008, 9.9 ± 0.3 ppt, plume average 10.6 ± 0.3 ppt) and estimated a global boreal fire source of 0.23 ± 0.19 kilotonnes (kt) yr$^{-1}$.*" Of the references given in WMO 2014 (Chapter 1, page 1.38), this is the only one that reported CH$_2$ClCH$_2$Cl.

We have also changed the word "*exclusively*" to "*predominantly*" when referring to anthropogenic sources of CH$_2$ClCH$_2$Cl.

**I.334: "Production has increased rapidly …": Can you give a reference for this increase in production?**

Reference added (same as in the next sentence)

**l. 362: Does the superscript "1" have any meaning? Footnote?**

The superscript was actually a missing reference, which has now been added (Carpenter and Reimann et al. 2015).

**l. 367: "2"-> "two"**

Done

**Table 1: Why not use the IATA code "FRA" for Frankfurt (rather than "FFT", which is the IATA code for Frankfort, Kentucky)?**

Done

**Table 1: Why is the sum of VSLS-CL excluding CH2ClCH2Cl not given for the other data for comparison?**

The sum of Cl-VSLS excluding $CH_2ClCH_2Cl$ is not a widely used number so it was not included in the Table apart from where necessary. There is no equivalent number reported in WMO. However, we have added the information for the CARIBIC flights between Frankfurt and Bangkok for comparison as suggested.

**Reviewer 2**

**In general, all the information on the samples: time, locations for all the data, number of flights for CARIBIC data, number of sampling at the ground-based stations should be added in this section. For instance, the 7 IAGOS-CARIBIC flights time should be mentioned. Days and months of samples should be specified in this section as well. It will help the reader to get the general feature of the sampling.**

**The altitude of CARIBIC needs to be shown. Have you filtered IAGOS-CARIBIC data to analyze data between 10 and 12 km only? The statistics of the sampling in this layer is needed.**

We have added more detail at the beginning of the methods section:

*"A total of 21 samples were collected at Hengchun between 7 March and 5 April 2013 with a further 22 samples taken at Cape Fuguei between 11 March and 4 April 2014. 28 samples were collected at Bachok between 20 January and 5 February 2014, during the period of the NE winter monsoon. The approximate location of each surface site is shown in Figure 1. The CARIBIC aircraft samples were collected during seven return flights between (i) Frankfurt (Germany) and Bangkok (Thailand), and (ii) Bangkok and Kuala Lumpur (Malaysia) during the periods December 2012 - March 2013 (4 flights) and November 2013 - January 2014 (3 flights). All CARIBIC flights in this region between December 2012 and January 2014 have been included in this analysis. With the exception of 3 samples that were taken at altitudes between 8.5 and 9.8 km, the CARIBIC samples were all (n=179) collected at altitudes between 10 and 12.3 km."*

The full CARIBIC flight dates have also been added to Figure 5.

**In the text it is mentioned that 10-12km over East Asia is the lower boundary of the TTL. It would be very helpful to show a map of TTL or a figure of TTL and aircraft altitudes together with respect of the flight tracks (latitude). It would be also useful to directly refer to Box 1-3, Figure 1 of Carpenter and Reimann et al. (2015) that shows the altitude range of the TTL.**

We have added a reference to the Figure from Carpenter and Reimann et al. (2015). We do not think it is necessary to reproduce a similar Figure here.

**About the results shown in Figure 3, a sentence explaining that three days have been chosen out of the seven days of the cold surge event would be helpful. The term "cold surge" should be mentioned**.

New text added: "*Three examples during this cold surge event are shown in Fig. 3 (b-d).*"

**In general, "see in supplement" is largely used in the manuscript but I would suggest to refer to figure number and section names of the supplement materials to help the reader.**

Done

**973**  **Results from Carpenter and Reimann et al., 2015 are cited as reference for Cl-VSLS**
**974**  **last information. For results at the lower TTL, it would be useful to recall the type of**
**975**  **observations used in the assessment report: aircraft campaigns and balloons.**

**976**  We have added the following sentence to Table 1: "*Data from the TTL was derived from*
**977**  *various aircraft and balloon campaigns.*"

**978**  **Table: Units need to be added in Table 1 and its caption**

**979**  Done

**980**  **In general, the way to write Cl-VSLS should be consistent along the captions and the**
**981**  **text (sometimes chlorinated VSLS or VSLS-Cl).**

**982**  Done

**983**  **Figure 1: "(red crosses)" for surface sites on the map need to be added in the caption**
**984**  **to guide the reader.**

**985**  Done

**986**  **Figure 2: We don't see the blue circle on the map.**

**987**  We apologise that the incorrect Figure was included in the original submission. The blue
**988**  circles have now been added.

**989**  **Figures 2,3,5: Helpful to have a recall of ground-based stations location.**

**990**  Blue circles have been added to Figures 2 and 3 showing the location of the surface
**991**  sampling sites.

**992**  **Figure 2 and 3: I would rather use letter to name the panels in the caption and I would**
**993**  **rather use numbers to link plots in upper panel with maps of the bottom panels.**

**994**  We use letters to label the arrows which refer to the individual NAME plots underneath.
**995**  Mixing letters and numbers would, in our opinion, be more confusing.

**996**  **Figure 3: Use arrows as for Figure 2 to help the reader to find the days that are**
**997**  **chosen for air masses origin (map below). In the caption it is mentioned "true**
**998**  **background levels", how these levels are estimated? Figure 3 a) is not specified, "a)"**
**999**  **should be added on the figure.**

**1000**  Arrows have been added to Figure 3 as requested. The baseline levels are taken from WMO
**1001**  2015, based on tropical mean background levels. The actual expected background is difficult
**1002**  to define for VSLS as they would be expected to vary substantially across the globe (i.e. with
**1003**  latitude and with distance from source). We have added a reference to WMO in the Figure
**1004**  caption.

**1005**  The missing (a) has been added to Figure 3.

**1006**  **Figure 4 and S2: What does CO anomaly mean? What is the reference value?**

**1007**  By the term "CO anomaly" we mean the fraction of CO observed at Bachok from industrial
**1008**  emissions from regions north of 20N. This is explained in Section 3 (lines 285-288) and in
**1009**  the supplement. We have added a sentence in the caption for Figure 4 to remind readers.

**1010**  **Line 126: "Both in the western Pacific region and globally". It is not clear what**
**1011**  **globally means.**

We have removed the word "globally".

**Line 147: "Various time" needs to be specified (see general comments).**

More sampling information has been added (see earlier response to reviewer 2))

**Line 150: Change "CARIBIC aircraft" to "IAGOS-CARIBIC aircraft" as CARIBIC is part of the IAGOS program.**

Done

**Line 230: Change "shows the 2014 data …" to "shows data from CAPE Fuguei in the end of March, beginning of April 2014".**

Text has been changed to "*shows the March/April 2014 data…*". In addition the sampling dates have been defined more clearly in the text (see earlier comment of reviewer 2). The dates are also evident in the Figures.

**Line 237: Change "March/April 2013" to "mid of March/beginning of April 2013".**

To be consistent with the text in the previous comment we have not made this change but note that the sampling dates have been defined more clearly in the text (see above). The dates are also evident on the Figures.

**Line 246: "January/February": the entire months are not shown so "end of January/beginning of February" would be more appropriate. "During this phase of the monsoon": A recall about the Asian Monsoon phases and references would be useful, maybe recall that it is the East Asian winter monsoon circulation as mentioned in the introduction.**

Text has been changed to "*late January/ early February 2014*". Sampling dates have been clarified in methods section.

We have added the words "*East Asian*" when referring to the monsoon and reminded readers that this was described earlier.

**Line 252: change "often" to "most of the case studies we are analyzing here" or "for N days out of Ntotal days of observations" or give the information in Line 277:**

We would prefer to keep the word "often". This was deliberately vague as we have not done any specific analysis to determine how many times the air may have picked up emissions from Taiwan. The observation was also based on the NAME animations which are referred to in line 285.

**Change "(see supplement for further details)" to "(Fig. S2 in the supplement material)".**

Done

**Line 287: Change "in all CARIBIC flights" to "in the seven CARIBIC flights" and remove "(7)".**

Done

**Line 334: Need a reference.**

Reference added

**Line 362: Need a reference.**

Reference added

**Line 383: Examples of other chemical pollutants would be useful**.

The pollution we were referring to are chemicals that are routinely found in industrialised
countries. These include CO, $O_3$, $CH_4$, volatile organic compounds (VOCs) including
hydrocarbons, oxygenated hydrocarbons and certain halocarbons. These measurements are
likely to feature in a future publications.

**Line 384: Remove "etc".**

Done